# Cerebrovascular Reactivity Assessment during Carbon Dioxide Inhalation Using SPECT

Yeong-Bae Lee [1,2] and Chang-Ki Kang [2,3,*]

1 Department of Neurology, Gil Medical Center, Gachon University College of Medicine, Incheon 21565, Korea; yeongbaelee@gachon.ac.kr
2 Neuroscience Research Institute, Gachon University, Incheon 21565, Korea
3 Department of Radiological Science, College of Health Science, Gachon University, Incheon 21936, Korea
* Correspondence: ckkang@gachon.ac.kr; Tel.: +82-32-820-4110

**Abstract:** Background: Perfusion single-photon emission computed tomography (SPECT) using an acetazolamide is an important clinical tool used to assess cerebrovascular reactivity (CVR) in patients, but its use has been limited to clinical diagnostics. This study aimed to preliminarily evaluate the feasibility of perfusion SPECT using carbon dioxide ($CO_2$). Methods: Ten healthy subjects participated in two consecutive SPECT scans using $CO_2$ inhalation. To evaluate brain perfusion after preprocessing, the voxel-by-voxel CVR values were averaged in 13 subgroup regions of interest (ROIs) based on a template. Subsequently, averaged CVR values of each ROI were analyzed based on both cerebellar hemispheres. Results: CVR values in the eight subgroup ROIs, which included vermis, both insula/cingulate, and frontal cortices, showed significant changes ($p < 0.05$). CVR values were higher in vermis and right insula/cingulate by 3.34% and 3.15%, respectively. Conclusions: This study showed that quantitative SPECT with $CO_2$ inhalation could be used to evaluate the voxel-based CVR in healthy subjects, which could be beneficial for elucidating induced hypercapnic states and for longitudinally investigating the healthy aging in brain vessels. Furthermore, the cerebrovascular hemodynamic parameters induced by $CO_2$ could play an important role as a biomarker to evaluate treatment progress in patients with cerebrovascular disease.

**Keywords:** single-photon-emission computed tomography; SPECT; carbon dioxide; $CO_2$; cerebrovascular reactivity

## 1. Introduction

Brain perfusion imaging using technetium 99m-labeled hexamethyl-propyleneamine-oxime (99mTc-HMPAO) single-photon emission computed tomography (SPECT) is used as a standard technique for evaluating and diagnosing cerebrovascular reactivity (CVR) in patients with cerebrovascular disease (CVD) [1,2]. The rapid circulation of 99mTc-HMPAO passes entirely through the blood–brain barrier (BBB) and is converted into hydrophilic compounds that rarely return to the bloodstream [1,3]. Since the distribution pattern shows perfusion upon administration, this phenomenon occurs within a few minutes of intravenous 99mTc-HMPAO administration and is retained in the brain for hours [3]. Therefore, the extra accumulated absorption of 99mTc-HMPAO into the brain tissue can be calculated by subtracting it in a subsequent SPECT image. These properties apply to any condition that causes brain perfusion within a short time, such as carbon dioxide ($CO_2$) inhalation.

Acetazolamide (ACZ) challenge via 99mTc-HMPAO SPECT has been commonly used as a clinical tool to evaluate various CVDs and estimate CVR [4,5]. Despite the common clinical use of this test, ACZ injections can sometimes cause side effects in the human body. $CO_2$, however, is a potent vasodilator of cerebral blood vessels that induces a hypercapnic state in the brain, and regional cerebral blood flow (rCBF) reactivity to $CO_2$ is one of the representative parameters of CVR assessment when using brain perfusion imaging [6,7].

The challenge of CVR to $CO_2$ has been investigated in various CVD patients and healthy subjects following $CO_2$ inhalation, and rCBF has been shown to increase significantly through various methods when $CO_2$ is inhaled [7,8]. Therefore, inhalation of air with an appropriate $CO_2$ concentration during a $^{99m}$Tc-HMPAO SPECT may be a more valuable and relatively low-cost clinical technique compared to ACZ injection.

Previously, many techniques for CVR have been developed to quantify changes in blood flow, especially in the middle cerebral artery (MCA), through vasodilatory challenges such as hypercapnia, apnea, or ACZ administration [8,9]. In the clinical setting, the use of transcranial Doppler (TCD), positron emission tomography (PET), and blood oxygen level-dependent (BOLD) functional magnetic resonance imaging (fMRI) have been applied to evaluate CVR in healthy volunteers and patients [9–11]. Among others, a BOLD fMRI performed with $CO_2$ inhalation as a task condition could potentially create a hypercapnic state that could predict both normal and abnormal hemispherical CVR [12,13]. Furthermore, various stimuli for vasodilation have been used to induce CVR, and several technologies for CVR have been variously introduced with suitable solutions [7,14]. Recently, a study was performed to conduct an effective diagnosis using multimodal imaging technologies including computer CT, MRI, and SPECT [15].

In contrast, SPECT for CVR evaluation has been considered a potential tool for studying brain function in vivo to determine CVD progression by identifying the magnitude and extent of damage to cerebral vascular flow and for predicting outcomes [16,17]. The presence and the degree of vasodilation associated with CVR have been significant prognostic indicators in chronic CVD patients with respect to perfusion SPECT. Although it is important to assess CVR using SPECT for a precise diagnosis, there are unresolved methodological concerns, for example, the qualitative and subjective assessment of regional CVR, which is significantly dependent on the examiner [18].

The purpose of this study was to determine the possible use of consecutive $^{99m}$Tc-HMPAO SPECT imaging with 5% $CO_2$ inhalation to evaluate CVR in healthy subjects and to calculate the voxel-based CVR values based on the automated anatomical labeling (AAL) template. Furthermore, we aimed to demonstrate the effects of $CO_2$ on a healthy human brain by performing a subgroup region of interest (ROI) analysis using an automated program developed for the quantitative evaluation of CVR.

## 2. Materials and Methods

### 2.1. Subjects and SPECT Acquisition and Reconstruction

Ten healthy right-handed subjects (male:female = 7:3; age (mean $\pm$ standard deviation (SD)) = 23.8 $\pm$ 2.10 years) voluntarily participated in this study (see Table 1) after it was approved by the institutional review board (IRB) in Gil Medical Center, Gachon University College of Medicine (IRB number: GDIRB2018-308), and written informed consent was signed. None of the participants had any history of cerebrovascular, neurological, neuropsychiatric, or neuropsychological diseases that involved cognitive decline. The procedure was carried out in accordance with the approved guidelines.

A one-day protocol of two consecutive SPECT (1st and 2nd) images with the administration of $^{99m}$Tc-HMPAO was conducted (Figure 1). Ten minutes after the first 20 mCi $^{99m}$Tc-HMPAO injection, a 1st SPECT was performed for an imaging duration of 15 min. When the 1st SPECT was completed, a gas mixture containing about 5% $CO_2$ was administered for 15 min through the subject's nose at a flow rate of 13–15 L per minute, which was controlled by a gas regulator. The 2nd 40 mCi $^{99m}$Tc-HMPAO for the 2nd SPECT was injected five minutes after the 1st SPECT was completed. Ten minutes after the 2nd $^{99m}$Tc-HMPAO injection, the 2nd SPECT was performed for the same imaging duration as the 1st SPECT. SPECT imaging was performed using a dual-headed rotating gamma camera system (Symbia T16, Siemens Medical Solutions, Erlangen, Germany) equipped with a low-energy, high-resolution (LEHR) parallel-hole collimator. The projection data were collected with 32 views per camera head in a 128 $\times$ 128 matrix at 25 s per angle with

a zoom factor of 1.45 in the step-and-shoot mode. This one-day protocol study took a total of about 55 min.

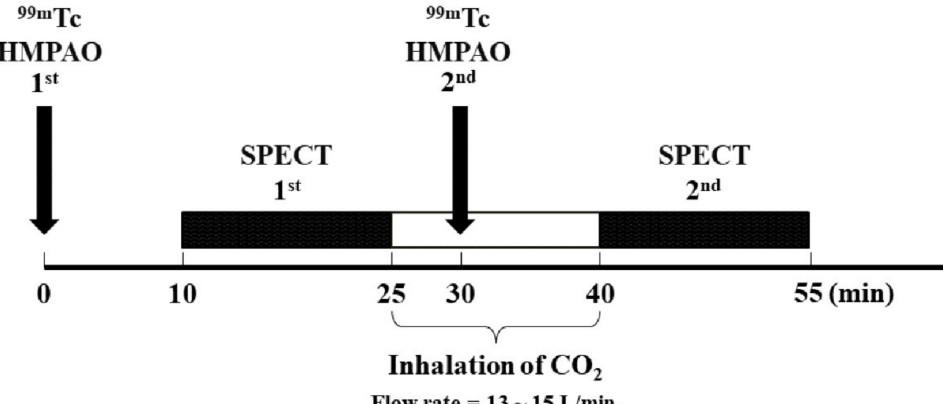

**Figure 1.** The acquisition protocol for the same day (one-day) consecutive first (20 mCi) and second (40 mCi) technetium 99m-labeled hexamethyl-propyleneamine-oxime ($^{99m}$Tc-HMPAO) single-photon emission computed tomography (SPECT) imaging. A gas mixture including about 5% $CO_2$ gas was administered through each subject's nose. The flow rate, controlled by a gas regulator, was 13–15 L per minute during approximately 15 min. Each scan time was 15 min (25 s × 32 steps, step-and-shoot mode) for each first and second SPECT.

These SPECT images were reconstructed on a 128 × 128 matrix using Flash (Siemens) three-dimensional iterative reconstruction (eight iterations and four subsets) after Gaussian-filtering (full width at half maximum (FWHM) = 6.6 mm). After reconstruction of the trans-axial tomographic images parallel to the orbitomeatal line, attenuation correction using Chang's method [19] was performed for each slice with a uniform attenuation coefficient of 0.11 cm, as shown in Figure 2.

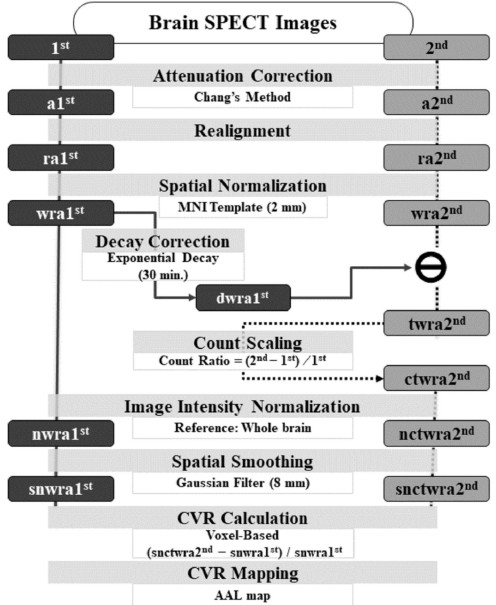

**Figure 2.** Diagram representing SPECT cerebrovascular reactivity (CVR) processing using the first and second SPECT data.



**Table 1.** Demographic information (N = 10).

| Number | Sex | Age | Pre-Scan | | | | Post-Scan | | | | Count | | |
|---|---|---|---|---|---|---|---|---|---|---|---|---|---|
| | | | SBP | DBP | MAP | HR | SBP | DBP | MAP | HR | 1st Scan | 2nd Scan | Ratio |
| #01 | M | 25 | 116 | 59 | 78 | 87 | 125 | 95 | 105 | 84 | 3593 | 10,599 | 2.95 |
| #02 | M | 23 | 116 | 65 | 82 | 60 | 131 | 67 | 88.33 | 67 | 3854 | 11,287 | 2.93 |
| #03 | F | 22 | 99 | 56 | 70.33 | 65 | 114 | 66 | 82 | 71 | 5223 | 15,440 | 2.96 |
| #04 | M | 23 | 119 | 60 | 79.67 | 60 | 122 | 67 | 85.33 | 56 | 3555 | 12,272 | 3.45 |
| #05 | M | 25 | 134 | 82 | 99.33 | 70 | 136 | 90 | 105.33 | 77 | 3543 | 10,453 | 2.95 |
| #06 | M | 26 | 143 | 69 | 93.67 | 61 | 153 | 66 | 95 | 62 | 3390 | 10,279 | 3.03 |
| #07 | M | 27 | 123 | 73 | 89.67 | 63 | 122 | 75 | 90.67 | 62 | 4222 | 12,963 | 3.07 |
| #08 | F | 21 | 91 | 52 | 65 | 51 | 98 | 60 | 72.67 | 61 | 3486 | 12,876 | 3.69 |
| #09 | F | 21 | 94 | 60 | 71.33 | 70 | 96 | 57 | 70 | 75 | 4441 | 13,696 | 3.08 |
| #10 | M | 25 | 123 | 67 | 85.67 | 65 | 130 | 76 | 94 | 66 | 4335 | 13,327 | 3.07 |
| Mean ± SD | | 23.80 ± 2.10 | 115.80 ± 16.86 | 64.30 ± 8.85 | 81.47 ± 10.89 | 65.20 ± 9.43 | 122.70 ± 17.07 * | 71.90 ± 12.3 * | 88.83 ± 11.92 * | 68.10 ± 8.60 | 3964.20 ± 583.96 | 12,319.20 ± 1668.23 | 3.12 ± 0.25 |

Abbreviations: SBP, systolic blood pressure; DBP, diastolic blood pressure; MAP, mean arterial pressure; HR, heart rate; SD, standard deviation. * $p < 0.05$ and it was analyzed with non-parametric Wilcoxon signed-rank test.

*2.2. SPECT CVR Processing*

A quantitative analysis was performed using the attenuation corrected from the 1st and 2nd SPECT data with voxel-based calculations to generate a three-dimensional CVR map, which was performed using an automated processing pipeline comprising the following preprocessing steps: realignment, spatial normalization, decay correction (subtraction), count scaling, image intensity normalization, and spatial smoothing (Figure 2). In particular, the realignment, spatial normalization, and spatial smoothing steps were processed using algorithms in the Statistical Parametric Mapping (SPM) software package (Wellcome Trust Centre for Neuroimaging). In detail, first, since two images were acquired per day, image displacement may have occurred due to the movement of the subject's head. To correct this, realignment was performed to estimate the rigid-body transformation of the 2nd SPECT image from the 1st SPECT image. Second, since all subjects had brains of different sizes, spatial normalization was performed to normalize the realigned SPECT image to the Montreal Neurological Institute (MNI) template space at 2 mm resolution. $^{99m}$Tc-HMPAO was injected twice in total so that the first-injected $^{99m}$Tc-HMPAO remained at the time of the second injection, causing a problem of reducing the sensitivity of the 2nd SPECT image. Third, therefore, the spatial-normalized image from the 1st SPECT image underwent radioactive decay signal correction (decay correction) to measure the amount of the 1st SPECT image signal. Then, the corrected image was obtained by subtracting the decay-corrected 1st SPECT image from the spatial-normalized 2nd SPECT image (subtraction) and correcting the signal intensity of the 2nd SPECT image due to residual radiation of the 1st SPECT. Fourth, count scaling was used to divide the signal intensity in each voxel in the spatial-normalized 2nd SPECT image by the ratio of radiation counts in each SPECT. Then, to solve the problem where $^{99m}$Tc-HMPAO affects the increase or decrease in blood flow in the brain, the 1st and 2nd SPECT data were normalized to the average signal strength of the brain (image intensity normalization). Finally, the intensity-normalized SPECT data were smoothed with an isotropic three-dimensional Gaussian kernel of 8 mm in FWHM so that the error distribution of the given data was smoothed sufficiently for statistical inference (spatial smoothing).

The CVR value (%) was subsequently calculated from each voxel in each SPECT image as shown in Equation (1) and then estimated to generate a three-dimensional voxel-wise CVR map. The formula was described as follows:

$$CVR \ (\%) = [(2nd - 1st)/1st] \times 100 \tag{1}$$

The positive CVR value could be derived from the site of normal perfusion pressure following the administration of a vasodilator such as $CO_2$, whereas the negative CVR value could be derived from the site where blood flow decreased. The AAL template consisted of 116 ROIs covering cortical and subcortical structures of the entire brain (54 symmetrical ROIs for each hemisphere including the frontal, temporal, parietal, and occipital cortices; the insula; striatum; thalamus; cerebellum; and 8 vermis ROIs in the medial plane). This template has been recently used for SPECT analysis studies [20,21]. We reconstructed a voxel-based CVR map from the voxel-wise CVR map (Figure 3) to average the CVR values in all voxels in each ROI. Furthermore, 13 subgroup CVR ROIs (ROI #01: left temporal cortex; ROI #02: right temporal cortex; ROI #03: vermis, medial; ROI #04: left insula/cingulate; ROI #05: right insula/cingulate; ROI #06: left frontal cortex; ROI #07: right frontal cortex; ROI #08: left occipital cortex; ROI #09: right occipital cortex; ROI #10: left parietal cortex; ROI #11: right parietal cortex; ROI #12: left central cortex; and ROI #13: right central cortex) were obtained with the voxel averages for all subjects (Figure 4 and Table 2). The mean CVR value in the cerebellum was then calculated to obtain relative CVR values in the 13 subgroup ROIs.

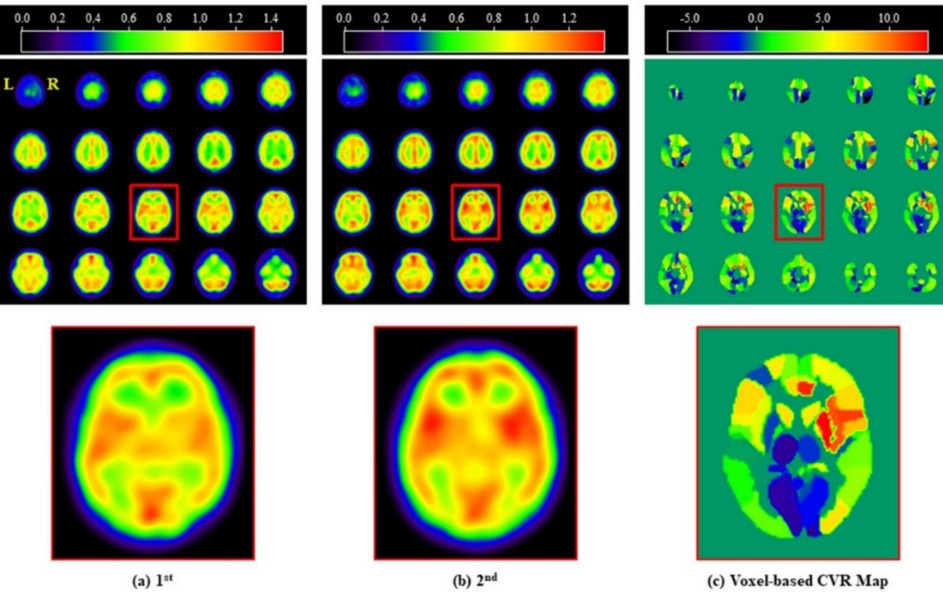

**Figure 3.** Standardized display supporting the assessment of the SPECT images obtained, including (**a**) first, (**b**) second, and (**c**) voxel-based CVR Map of a representative subject. The colored bars indicate the signal intensity in both (**a**) and (**b**), and the CVR (%) value in (**c**), respectively. Abbreviations: L, left; R, right.

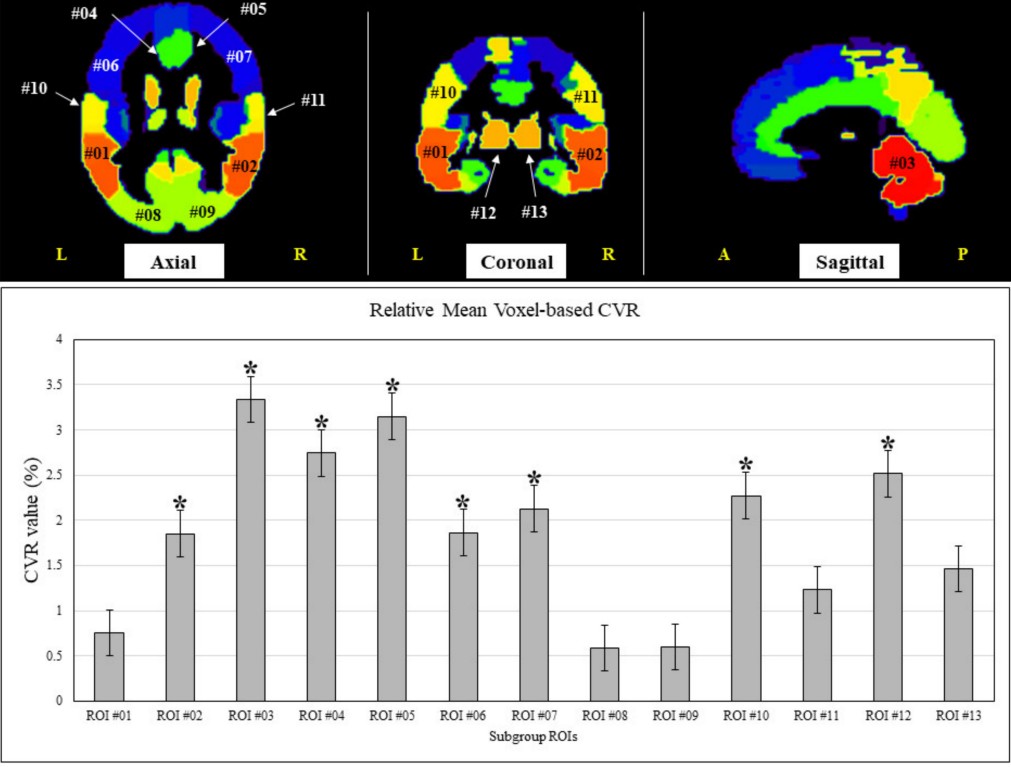

**Figure 4.** Graph of relative mean voxel-based CVR across 10 subjects in each of the 13 subgroup regions of interest (ROIs) according to a mean reference CVR value. ROI #01: left temporal cortex; ROI #02: right temporal cortex; ROI #03: vermis, medial; ROI #04: left insula/cingulate; ROI #05: right insula/cingulate; ROI #06: left frontal cortex; ROI #07: right frontal cortex; ROI #08: left occipital cortex; ROI #09: right occipital cortex; ROI #10: left parietal cortex; ROI #11: right parietal cortex; ROI #12: left central cortex; and ROI #13: right central cortex (see Table 2). Abbreviations: L, left; R, right; A, anterior; P, posterior; * indicates statistically significance ($p < 0.05$).

**Table 2.** The relative mean voxel-based CVR values and statistical *p*-values for non-parametric one-sample Wilcoxon t-test with a mean reference CVR value of −2.28% in 13 subgroup ROIs.

| ROI Number | Region Name | Hemisphere | # of ROIs | Relative Mean Voxel Based CVR Value (%) | *p*-Value |
|---|---|---|---|---|---|
| ROI #01 | Temporal | L | 10 | 0.756 | 0.285 |
| ROI #02 | Temporal | R | 10 | 1.849 | 0.047 * |
| ROI #03 | Vermis | M | 8 | 3.337 | 0.007 * |
| ROI #04 | Insula, Cingulate | L | 4 | 2.745 | 0.005 * |
| ROI #05 | Insula, Cingulate | R | 4 | 3.148 | 0.007 * |
| ROI #06 | Frontal | L | 15 | 1.862 | 0.005 * |
| ROI #07 | Frontal | R | 15 | 2.127 | 0.007 * |
| ROI #08 | Occipital | L | 6 | 0.586 | 0.508 |
| ROI #09 | Occipital | R | 6 | 0.597 | 0.285 |
| ROI #10 | Parietal | L | 6 | 2.274 | 0.017 * |
| ROI #11 | Parietal | R | 6 | 1.231 | 0.059 |
| ROI #12 | Central | L | 4 | 2.515 | 0.007 * |
| ROI #13 | Central | R | 4 | 1.463 | 0.285 |

Abbreviations: L, left; R, right; M, medial; ROI, region of interest. * $p < 0.05$.

### 2.3. Statistical Analysis

We performed statistical analyses using a statistical analysis package (IBM SPSS, version 25). The assumption of normality was tested using the Shapiro–Wilk test. The non-parametric Wilcoxon signed-rank test was used to test any difference in systolic and diastolic blood pressure, mean arterial pressure (MAP), and heart rate (HR) between pre- and post-scan measurements. To test the differences in all subgroup ROIs on a test value derived from the CVR values in the cerebellum region, a one-sample Wilcoxon signed-rank test was performed. A *p*-value of 0.05 was considered significant for the statistical analysis.

### 3. Results

The demographics and experimental results for all subjects are shown in Table 1. The MAP of all subjects ranged from 65–99.33 mmHg at pre-scan and 70–105.33 mmHg at post-scan. The MAP differed significantly from pre-scan to post-scan ($p = 0.011$), probably due to the hypercapnic stimulation via $CO_2$ inhalation; however, the HR did not ($p = 0.092$). The maximum ratio of radiation counts was 3.69 and the minimum was 2.93 (Table 1). The mean (SD) end-tidal $CO_2$ (Et-$CO_2$) values were 26.96 (2.56) mmHg and 43.63 (2.48) mmHg before and during $CO_2$ inhalation (N = 7), respectively.

The pre-processed first and second SPECT images of a representative subject are shown in Figure 3. In the second image, there was an increase in signal intensity in the frontal, temporal, and occipital cortices and the cerebellum compared to the first image. However, there were no specific values demonstrating increases in target ROIs between the first and second images. As shown in Figure 3c, there was a quantitative difference between the two SPECT images in a total of 116 ROIs on the voxel-based CVR map. The mean CVR value in the cerebellum was −2.28%, which was used for the relative CVR values in subgroup ROIs.

As shown in Table 2 and Figure 4, the CVR values in the following eight subgroup ROIs were significant: ROI #02 (right temporal cortex), ROI #03 (vermis, medial), ROI #04 (left insula/cingulate), ROI #05 (right insula/cingulate), ROI #06 (left frontal cortex), ROI #07 (right frontal cortex), ROI #10 (left parietal cortex), and ROI #12 (left central cortex) ($p < 0.05$). Specifically, the high relative mean CVR values were 3.337% in ROI #03 (vermis) and 3.148% in ROI #05 (right insula/cingulate). The low relative CVR values were 0.586% in ROI #08 (left occipital cortex) and 0.597% in ROI #09 (right occipital cortex), but they were not statistically significant.

## 4. Discussion

In this study, we assessed the feasibility of measuring CVR using a [99m]Tc-HMPAO SPECT with 5% $CO_2$ inhalation in healthy subjects. Unlike previous studies, our voxel-wise CVR map was constructed based on an AAL brain template with 116 ROIs [20,22]. An automated processing program quantitatively measured the mean CVR value of voxels in each subgroup ROI to elucidate the effect of the hypercapnic state on a healthy human brain. We found that this quantitative CVR analysis of the subgroup ROIs could be used as an exploratory approach to assess vascular reactivity in healthy subjects [23,24].

The findings demonstrated that $CO_2$ perfusion SPECT could provide evidence for the relationship between the reactive effect of $CO_2$ and CBF in specific vascular territories. Using the CVR maps obtained from all the subjects, the relative CVR values in eight subgroup ROIs, including the right temporal cortex, vermis, left and right insula/cingulate, left and right frontal cortex, and left parietal and central cortex, showed significant changes ($p < 0.05$) relative to the CVR value in the cerebellum. The results showed high relative mean CVR values in the right and left insula/cingulate cortex (ROI #04 and #05, respectively) and in the cortical areas other than the vermis (ROI #03), and low relative mean CVR values in the right and left occipital cortex (ROI #08 and #09, respectively). According to previous studies, an increase in $CO_2$ concentration in inspired air is known to cause vascular changes in the brain, including increased CBF and higher $CO_2$ and $O_2$ concentrations in the blood [25]. In the present study, the insula/cingulate cortex (anterior vascular territory), which receives blood through the anterior cerebral artery (ACA), and the occipital cortex (posterior vascular territory), which receives blood through the posterior cerebral artery (PCA), did not show significant changes like the cerebellum. This suggests that $CO_2$ has various effects on vascular territories, which has been shown in several studies in which major cerebral arteries, such as the MCA, PCA, and ACA, undergo some changes after $CO_2$ is applied in patients and healthy subjects [26,27].

In contrast to previous studies investigating the feasibility of CVR measurements using $CO_2$ on patients, few studies using SPECT and $CO_2$ inhalation have been conducted on healthy subjects [8,28]. $CO_2$ has fewer side effects than ACZ, is more likely to cause blood flow changes, which allows researchers to evaluate CVR on healthy subjects as well as patients, and is commonly used in BOLD fMRI studies [12,13]. Furthermore, although SPECT is a standard tool for perfusion imaging in the clinical diagnosis of patients, it is not sufficiently used for quantitative evaluation and objective diagnosis. Therefore, the proposed quantitative CVR mapping technique in this study could provide the basis for the development of objective and clinical diagnostic methods in future SPECT CVR studies and for a direct comparison between SPECT and fMRI CVR measurements. Furthermore, this technique can be used as a medical check-up tool by analyzing the degree of change through follow-up tests which primarily prevent cerebrovascular disease, and it can be used to predict the penumbra area of patients with acute cerebral infarction, and to determine the degree of autoregulation damage or recovery of cerebral vessels [29]. Lastly, it can help determine the proper timing of vascular bypass operation for moyamoya disease, a rare cerebrovascular disease, without any vasodilation drug [30].

However, this study has some limitations. First, only a small number of young, healthy volunteers were included. Moreover, gender inequality with small groups can make it difficult to draw definitive conclusions. Further studies should examine optimal $CO_2$ concentrations for providing reliable and reproducible CVR values [31]. Additionally, issues involved with the evaluation of therapeutic effects, follow-up assessments of patients, and/or vascular aging assessments using CVR measurements in healthy subjects via SPECT should be managed in future studies [32]. Nevertheless, follow-up assessments can be repetitively conducted with SPECT and $CO_2$ for the different vascular reactivity in several diseases such as vascular dementia or Alzheimer's dementia. Additionally, vascular reactivity according to the prevalence of hypertension, hyperlipidemia, and diabetes can be investigated and research on the auto-regulatory ability of blood vessels in the area of

cerebral infarction in patients with acute cerebral infarction can be conducted [33]. Further studies can confirm the association between ageing and changes in vascular capacity [34].

## 5. Conclusions

In conclusion, two consecutive $^{99m}$Tc-HMPAO SPECTs with 5% $CO_2$ inhalation is a simple and feasible method for assessing $CO_2$ reactivity as a substitute for ACZ, especially in healthy subjects and for clinical research purposes. The proposed quantitative CVR mapping technique for cerebral perfusion reserve may be an important basis for the development of objective and supplementary diagnostic methods for basic and clinical research. Importantly, this findings of this study regarding $CO_2$ could provide the basis for the easy use of $CO_2$ in perfusion studies on cerebrovascular ageing or various cerebral diseases on healthy people in the future. Although CVR research is actively underway in MRI, it is still limited in use as a clinical device for diagnosis. However, SPECT is used as a clinical device for diagnosis of cerebrovascular diseases, but the use of vasodilators is essential. Therefore, the purpose of this study was to find a way to complement the essential use of these contrast agents and maximize the effect of using SPECT.

**Author Contributions:** Conceptualization, C.-K.K. and Y.-B.L.; methodology, C.-K.K.; software, C.-K.K.; validation, C.-K.K. and Y.-B.L.; formal analysis, C.-K.K.; investigation, C.-K.K.; resources, C.-K.K.; data curation, C.-K.K.; writing—original draft preparation, writing—review and editing, visualization, supervision, project administration, and funding acquisition, C.-K.K. and Y.-B.L. All authors have read and agreed to the published version of the manuscript.

**Funding:** This work was supported by a grant of the Korea Health Technology R&D Project through the Korea Health Industry Development Institute (KHIDI) funded by the Ministry of Health & Welfare, Republic of Korea (No. HI17C0557) and the Bio & Medical Technology Development Program of the National Research Foundation (NRF) grant (2020M3A9E4104384).

**Institutional Review Board Statement:** The study was conducted according to the guidelines of the Declaration of Helsinki, and approved by the institutional review board (IRB) in Gil Medical Center, Gachon University College of Medicine (IRB number: GDIRB2018-308).

**Informed Consent Statement:** Informed consent was obtained from all subjects involved in the study.

**Data Availability Statement:** The data presented in this study are openly available in FigShare [dx.doi.org/10.6084/m9.figshare.13643324].

**Acknowledgments:** The authors thank all participants, especially the support staffs (M.G. Song and C.A. Park) in Gachon University.

**Conflicts of Interest:** The authors declare no conflict of interest.

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
