# Peer review of "Cerebrovascular Reactivity Assessment during Carbon Dioxide Inhalation Using SPECT"

_applsci, doi:10.3390/app11031161_

Round 1

Reviewer 1 Report

  1. Poor quality of figures 1 and 2. Please improve.
  2. Weak point of this work is that only 10 participants took part in the study with inequality of male and femal.
  3. Background to the study requires some amendments! Only 2 positions from 2020, one from 2019. It seems a little bit outdated.
  4. There is lack of section regarding further research plans, which would make this paper more interesting.
  5. Lack of section regarding study background of similar systems/solutions. 
  6. Overall a very good, solid work. Some amendments are however necessary.

Reviewer 2 Report

The study is interesting and its showing a high potential to be of use in neurology/neurosurgery. However, it is hard to see exactly how significant are the results for clinical practice, as this is not a case-control study (CO2 v ACZ, especially as in the  conclusions it is states that the method could be used as a substitute for ACZ, and as the study group is very low (only 10 subjects).

The clarity of the text could be improved, especially as the journal they have submitted the manuscript is a more general purpose journal, and not one specialized in imaging, making it difficult to follow for a non-specialist. 
